# Individual housing of male C57BL/6J mice after weaning impairs growth and predisposes for obesity

Lidewij Schipper[1,2]*, Steffen van Heijningen[2], Giorgio Karapetsas[2], Eline M. van der Beek[1,3], Gertjan van Dijk[2]

1 Danone Nutricia Research, Utrecht, The Netherlands, 2 GELIFES, Groningen Institute for Evolutionary Life Sciences, University of Groningen, Groningen, The Netherlands, 3 Department of Pediatrics, University Medical Center Groningen, University of Groningen, Groningen, The Netherlands

* lidewij.schipper@danone.com

**Data Availability Statement:** All raw data files are available from the data repository of the University of Groningen (https://hdl.handle.net/10411/PBYTMB).

## Abstract

For (metabolic) research models using mice, singly housing is widely used for practical purposes to study e.g. energy balance regulation and derangements herein. Mouse (social) housing practices could however influence study results by modulating (metabolic) health outcomes. To study the effects of the social housing condition, we assessed parameters for energy balance regulation and proneness to (diet induced) obesity in male C57Bl/6J mice that were housed individually or socially (in pairs) directly after weaning, both at standard ambient temperature of 21˚C. During adolescence, individually housed mice had reduced growth rate, while energy intake and energy expenditure were increased compared to socially housed counterparts. At 6 weeks of age, these mice had reduced lean body mass, but significantly higher white adipose tissue mass compared to socially housed mice, and higher UCP-1 mRNA expression in brown adipose tissue. During adulthood, body weight gain of individually housed animals exceeded that of socially housed mice, with elevations in both energy intake and expenditure. At 18 weeks of age, individually housed mice showed higher adiposity and higher mRNA expression of UCP-1 in inguinal white but not in brown adipose tissue. Exposure to an obesogenic diet starting at 6 weeks of age further amplified body weight gain and adipose tissue deposition and caused strong suppression of inguinal white adipose tissue mRNA UCP-1 expression. This study shows that post-weaning individual housing of male mice impairs adolescent growth and results in higher susceptibility to obesity in adulthood with putative roles for thermoregulation and/or affectiveness.

## Introduction

To study effects of environmental factors on obesity and to evaluate the effects of novel therapies to prevent or treat obesity in humans, the use of animal models remains indispensable. Rodent models for obesity share some of the characteristics of obesity in humans and include those that are based on genetic alterations and/or environmental manipulations such as

**Funding:** This study was funded by Danone Nutricia Research. The funder provided support to the current study by covering research costs and in the form of salaries for authors LS and EMvdB. The specific roles of these authors are articulated in the 'author contributions' section Author contributions: LS GvD conceived and designed the experiments. LS SvH GK performed the experiments and analyzed the data. LS SvH GK EMvdB GvD were involved in decision to publish and preparation of the manuscript.

**Competing interests:** Authors LS and EMvdB are employed by Danone Nutricia Research. This does not alter our adherence to PLOS ONE policies on sharing data and materials. Danone Nutricia Research declares no commercial interest in subject matter, methods or materials discussed in the manuscript. Authors SvH GK and GvD declare no conflict of interest.

exposure to high fat diet or chronic stress [1, 2]. Such manipulations typically modulate the central and peripheral regulation of energy balance, resulting in hyperphagia and increased adipose tissue deposition. Many studies apply individual housing of the animal (i.e. one animal per cage) to allow quantification of food intake, energy expenditure and other behavioural, metabolic or physiological parameters relevant to energy balance regulation at the individual level. For social species such as rodents however, social isolation that comes with individual housing may cause chronic stress [3, 4], which can lead to neurocognitive impairments and altered anxiety and depression-like behaviours [5–7]. Moreover, based on the results of a systematic review and meta-analysis we recently conducted [8], we concluded that individual versus social housing of rats and mice alters metabolic health status. For instance, individually housed mice show increased energy intake and altered adipose tissue deposition compared to socially housed mice, thought the extent to which this occurs may be sensitive to strain, diet and other environmental factors [9–12]. Social isolation stress may indeed affect the neurobiological control of food intake by affecting brain area's involved in energy balance regulation such as the hypothalamus [13, 14] and the reward system [15]. Among the mechanisms through which social isolation can induce these changes is an increase in hypothalamic-pituitary-adrenal (HPA) activity and responsiveness [16–19], affecting circulating glucocorticoid levels that can directly target the CNS-adipose tissue axis and favoring visceral fat storage [20–22]. For mice in particular, individual housing may also provide a direct physiological challenge with respect to thermoregulation. Social thermoregulation by huddling is a strategy applied by many rodent species to conserve energy [23]. The standard environmental temperature in laboratories (i.e. around 20° C) is below the thermoneutral zone defined for mice [24]. Increased energy intake, expenditure and/or energy storage could therefore be necessary to support the demands of increased thermogenesis when mice are housed individually.

Due to critical steps in brain and endocrine development that take place in the period between weaning and sexual maturation in rodents, adolescence represents a period of heightened vulnerability to social isolation stress [25]. Indeed, profound abnormalities in brain structural and functional development due to social isolation occur specifically during adolescence, disrupting cognitive and behavioural function at later life stages [26–28]. Adolescence is also a period of rapid lean body growth [29], which is accompanied by a high energy demand. Moreover, this period also represents a critical phase for the development and maturation of (visceral) white adipose tissue [30–32] and the structural and functional maturation of the hypothalamic circuits that control food intake [33]. Housing-induced changes in energy use and distribution during this critical stage of life may therefore interfere with adolescent growth and body composition development. This may have a strong and permanent impact on energy balance regulation and metabolic phenotype. The metabolic consequences of individual housing starting at adolescent age have only been investigated in a limited number of studies in mice. Moreover, these studies appear to be contradicting, for instance early life individual housing was reported to increase [34], reduce [35–37] or not to affect [11, 38–40] body at later (adult) life stage. Likewise, adult adiposity at 11 to 12 weeks was increased [34] or not affected [36] as a result of individual housing directly after weaning. However, the consequences of post-weaning individual housing on body composition at earlier stage of live, i.e. during adolescence, remain to be characterized. Importantly, whereas it is was proposed that individually housing induces adaptations in energy balance regulation in order to support increased energy demand for thermogenesis [8], there are no studies yet comparing energy intake and energy expenditure of mice housed individually or socially directly after weaning and in relation to (long term) metabolic phenotype. In the current study, we aimed to characterize the short- and long-term metabolic consequences of post-weaning individual housing for mice. We hypothesized that individual housing increases visceral body fat deposition. To this end, we

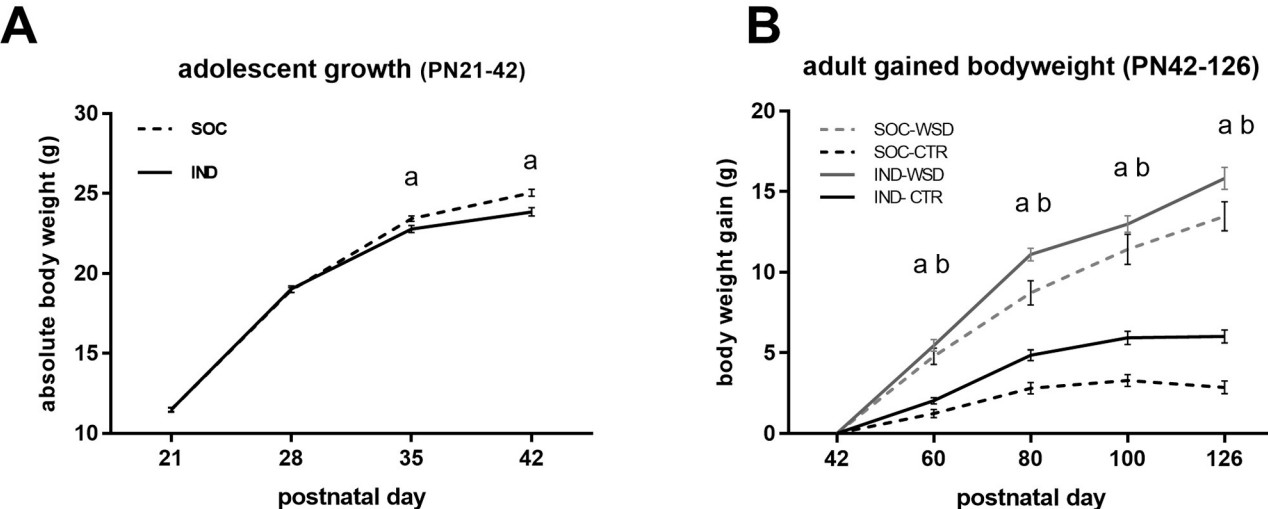

**Fig 1. Body weight (gain) of individually and socially housed mice.** (A) absolute body weight of adolescent mice. Groups did not differ in body weight at baseline (PN21; IND, *n = 28*, 11.5 ± 0.15; SOC, *n = 30*, 11.5 ± 0.12). Over the course of 3 weeks IND showed reduced body weight gain compared to SOC (p < 0.001). (B) Body weight gain during adulthood and either control diet (CTR) or western style diet (WSD) exposure. At baseline (PN42), absolute body weight of IND was lower than that of SOC (two-way ANOVA, main effect of housing condition: f (1,88) = 27.272, p < 0.001) while there were no differences between CTR and WSD exposed groups (IND-CTR, *n = 27*, 24.1 ± 0.27; IND-WSD, *n = 24*, 24.8 ± 0.30; SOC-CTR, *n = 23*, 26.2 ± 0.15; SOC-WSD, *n = 20*, 25.7 ± 0.36). Gained body weight between PN42 and PN126 was increased due to individual housing and WSD exposure (p < 0.001); Data are means ± SEM; [a] = significant (main) effect of housing (p < 0.05); [b] = significant main effect of diet (p < 0.05).

evaluated, at adolescent and adult age (diet induced) body weight (gain), body composition, and markers of energy balance regulation in male C57BL/6 mice housed individually (IND) or socially (in pairs; SOC) from weaning onwards.

## Results

### Individual housing of mice reduces adolescent growth rate while increasing adiposity

The growth rate of individually housed mice (IND) was reduced compared to that of socially housed mice (SOC) during adolescence (from weaning at postnatal day (PN) 21 until PN42; Fig 1A). The difference between IND and SOC in body weight first reached statistical significance at PN35. Upon dissection at PN43, IND showed a reduced femur length (Fig 2C) and—width (Table 1) compared to SOC while the reduction in lean body mass percentage (LBM%; Fig 2A) was not significant. Despite their lower bodyweight, IND showed higher adiposity than SOC at this age as evident from a higher body fat % (Fig 2B) and increased white adipose tissue (WAT) depots weights (Table 1). These differences in body composition were accompanied by higher plasma leptin of IND compared to that of SOC, however, there were no effects of housing situation on plasma levels of adiponectin, insulin and corticosterone (CORT) (Table 1).

### Individual housing of mice increases adult body weight gain and adiposity regardless of adult diet exposure

Between PN42 and PN126, IND and SOC were exposed to either a western style diet (WSD; 40En% as fat) or a control low fat diet (CTR, 20En% as fat). While absolute body weight of IND groups was significantly lower than that of the SOC groups at baseline (PN42) this

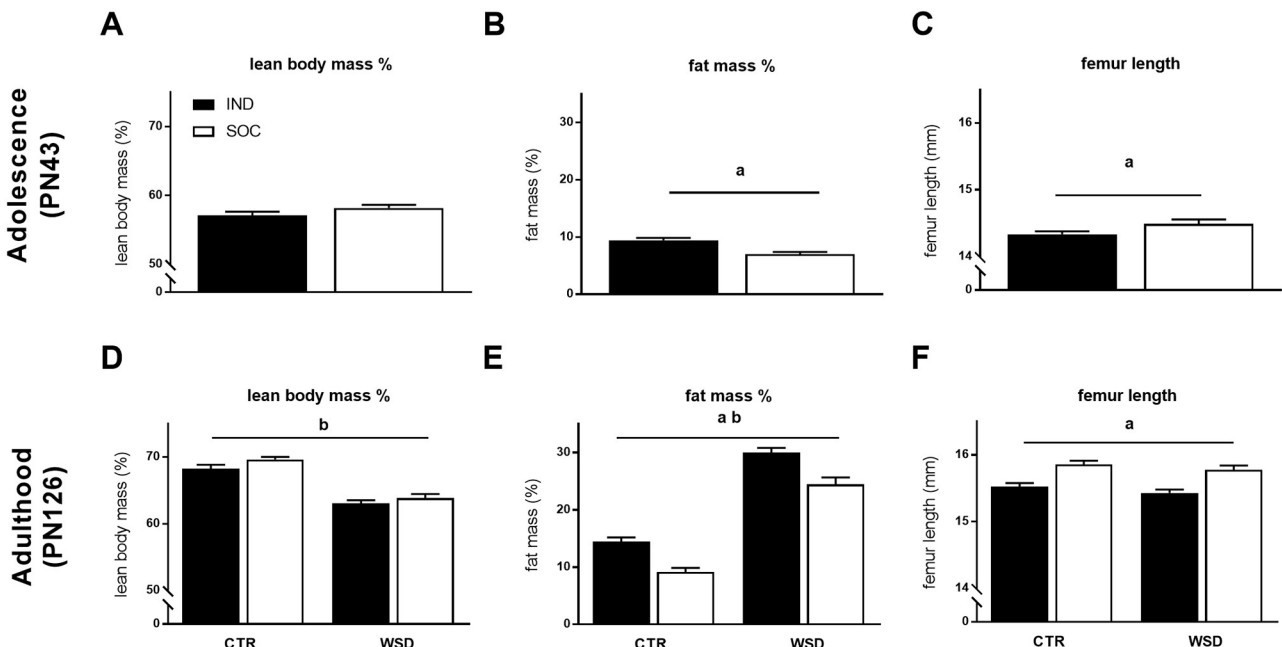

**Fig 2. Body composition of individually and socially housed mice.** (A) lean body mass %, (B) fat mass % and (C) femur length of adolescent (PN43) mice housed individually (IND, *n = 28*) and socially (SOC, *n = 30*); (D) lean body mass %, (E) fat mass % and (F) femur length of adult (PN126) mice housed individually or socially and exposed to either control diet (CTR) or western style diet (WSD) between PN42 and PN126 (IND CTR, *n = 27*; IND WSD, *n = 24*; SOC CTR, *n = 21*; SOC WSD, *n = 20*). Data are means ± SEM; [a] = significant (main) effect of housing (p < 0.05); [b] = significant main effect of diet (p < 0.05).

**Table 1. Absolute bodyweight, organ and tissue characteristics and plasma hormones of adolescent mice.**

|  | IND | SOC |  |
|---|---|---|---|
|  | *(n = 26–28)* | *(n = 29–30)* |  |
| Absolute body weight (g) | 23.85 ± 0.26 | 25.04 ± 0.22 | a |
| Carcass lean body mass (g) | 13.63 ± 0.22 | 14.57 ± 0.19 | a |
| Carcass fat mass (g) | 2.11 ± 0.09 | 1.62 ± 0.09 | a |
| organ / tissues |  |  |  |
| WAT,inguinal (mg) | 304.67 ± 15,20 | 259.67 ± 12.99 | a |
| WAT, epididymal (mg) | 395.26 ± 19.81 | 326.73 ± 18.76 | a |
| WAT, retriponeal (mg) | 80.56 ± 5.09 | 66.26 ± 4.18 | a |
| WAT, perirenal (mg) | 53.41 ± 3.03 | 42.46 ± 2.04 | a |
| BAT, interscapular (mg) | 143.25 ± 5.48 | 124.66 ± 4.38 | a |
| liver (g) | 1.04 ± 0.03 | 1.12 ± 0.04 |  |
| femur width (mm) | 1.28 ± 0.01 | 1.38 ± 0.01 | a |
| plasma hormones |  |  |  |
| adiponectin (mcg/ml) | 8.07 ± 0.27 | 7.97 ± 0.32 |  |
| leptin (ng/ml) | 2.92 ± 0.28 | 1.92 ± 0.23 | a |
| insulin (ng/ml) | 0.78 ± 0.12 | 0.94 ± 0.14 |  |
| CORT (ng/ml) | 33.86 ± 4.026 | 38.93 ± 4.11 |  |

Bodyweight, organ and tissue characteristics and plasma hormones of adolescent (PN43) mice housed individually (IND) or socially (SOC). Data are means ± SEM; [a] = significant effect of housing (p < 0.05).

**Table 2. Absolute bodyweight, organ and tissue characteristics and plasma hormones of adult mice.**

| | IND-CTR | IND-WSD | SOC-CTR | SOC-WSD | |
|---|---|---|---|---|---|
| | *(n = 27)* | *(n = 24)* | *(n = 21–23)* | *(n = 18–20)* | |
| Absolute body weight (g) | 30.14 ± 0.43 | 40.60 ± 0.83 | 29.07 ± 0.40 | 39.12 ± 1.09 | *a*, b |
| Carcass lean body mass (g) | 20.56 ± 0.29 | 25.16 ± 0.50 | 19.93 ± 0.41 | 24.97 ± 0.83 | b |
| Carcass fat mass (g) | 4.23 ± 0.27 | 11.87 ± 0.53 | 2.56 ± 0.25 | 9.41 ± 0.94 | a, b |
| organ / tissues | | | | | |
| WAT,inguinal (g) | 0.48 ± 0.03 | 1.45 ± 0.08 | 0.32 ± 0.02 | 1.10 ± 0.11 | a, b |
| WAT, epididymal (g) | 0.88 ± 0.05 | 2.50 ± 0.11 | 0.53 ± 0.06 | 2.00 ± 0.13 | a, b |
| WAT, retriponeal (mg) | 211.11 ± 16.42 | 649.65 ± 31.71 | 125.00 ± 17.10 | 587.78 ± 31.63 | a, b |
| WAT, perirenal (mg) | 103.70 ± 9.17 | 360.42 ± 25.28 | 68.18 ± 7.11 | 300.00 ± 33.26 | a, b |
| BAT,interscapular (mg) | 190.00 ± 9.00 | 422.50 ± 24.30 | 140.00 ± 7.53 | 335.00 ± 32.73 | a, b |
| liver (g) | 1.12 ± 0.04 | 1.69 ± 0.08 | 0.99 ± 0.05 | 1.63 ± 0.09 | a, b |
| femur width (mm) | 1.29 ± 0.01 | 1.26 ± 0.01 | 1.47 ± 0.02 | 1.46 ± 0.02 | a |
| plasma hormones | | | | | |
| adiponectin (mcg/ml) | 13.88 ± 1.13 | 13.20 ± 1.19 | 15.75 ± 1.45 | 10.88 ± 0.68 | b |
| leptin (ng/ml) | 3.36 ± 0.45 | 29.60 ± 5.27 | 1.32 ± 0.34 | 18.70 ± 2.61 | a, b |
| insulin (ng/ml) | 0.32 ± 0.04 | 1.48 ± 0.22 | 0.18 ± 0.03 | 0.99 ± 0.13 | a, b |
| CORT (ng/ml) | 44.59 ± 4.76 | 43.36 ± 5.86 | 62.29 ± 6.20 | 43.96 ± 5.05 | a, b |

Bodyweight, organ and tissue characteristics and plasma hormones of adolescent (PN126) mice housed individually (IND) or socially (SOC) and exposed to either control diet (CTR) or Western Style diet (WSD) between PN42 and PN126. Data are means ± SEM;

[a] = significant main effect of housing (p < 0.05);

*a* = trend (0.05 < p <0.1);

[b] = significant main effect of diet (p < 0.05).

difference was lost at PN60 (S1 Table). Due to the difference between groups at baseline, adult body weight gain was compared by using weight gain from baseline. Adult body weight gain of IND exceeded that of SOC (Fig 1B) and exposure to WSD also increased body weight gain (Fig 1B). At PN126, 12 weeks after the adult diet exposure, the absolute body weight of IND was higher than that of SOC, albeit not significantly, and WSD exposed mice had significantly higher body weight than mice exposed to CTR (Table 2). Moreover, individual housing as well as WSD exposure increased adiposity (Fig 2E and Table 2). While adult LBM% was reduced by WSD exposure it remained unaffected by housing conditions (Fig 2D). In contrast, femur length and width were reduced in IND compared to SOC, while adult diet did not influence femur characteristics (Fig 2E and Table 2). Adult leptin and insulin levels were increased by both individual housing and adult WSD exposure, while adiponectin was reduced by WSD exposure only. Plasma CORT tended to be increased in SOC compared to IND and blunted by adult WSD exposure, which appeared to be caused specifically by higher CORT levels in the SOC-CTR group compared to other groups, though the interaction between housing and adult diet did not reach significance (Table 2).

## Individual housing of mice increases energy expenditure and energy intake and results in anhedonia

Indirect calorimetry revealed that energy expenditure was increased in IND compared to SOC: During adolescence (PN40-40), IND showed a trend for higher energy expenditure during the light phase (P = 0.08), this effect reached significance after adjusting for LBM (Table 3). At adult age (PN106-108) IND also showed higher energy expenditure during the light phase,

**Table 3. Indirect calorimetry during adolescence.**

|  | IND | SOC |  |
|---|---|---|---|
|  | (*n = 26–28*) | (*n = 14–15*) |  |
| Energy Expenditure Light Phase |  |  |  |
| (kJ/g body weight) | 0.85 ± 0.01 | 0.82 ± 0.01 | *a* |
| (kJ/g LBM) | 1.49 ± 0.02 | 1.41 ± 0.02 | a |
| Energy expenditure Dark Phase |  |  |  |
| (kJ/g body weight) | 1.02 ± 0.02 | 1.00 ± 0.02 |  |
| (kJ/g LBM) | 1.80 ± 0.03 | 1.73 ± 0.02 |  |
| Cumulative energy intake (72 hour) |  |  |  |
| (kJ/ g body weight) | 10.05 ± 0.51 | 8.31 ± 0.37 | a |
| (kJ/g LBM) | 17.65 ± 0.88 | 14.29 ± 0.63 | a |

Average Energy Expenditure (kJ) during light (12 hr) and dark (12 hr) phase and cumulative energy intake of adolescent (PN40-PN42) mice housed individually (IND) or socially (SOC). Data are means *per cage* ± SEM;

[a] = significant effect of housing ($p < 0.05$).

[*a*] = trend ($0.05 < p < 0.1$).

while energy expenditure was increased in both the dark and light phase by WSD exposure (Table 4). At both adolescent and adult age, individual housing increased the deposition of interscapular brown adipose tissue (BAT; Tables 1 and 2) in which the brown adipocyte specific Uncoupling Protein 1 (UCP-1) mediates energy expenditure to support thermogenesis. Individual housing increased mRNA UCP-1 expression in BAT of adolescent animals and in inguinal WAT of adult animals, the latter suggesting increased browning of white adipose tissue due to individual housing. Exposure to WSD reduced UCP-1 expression in inguinal WAT (Fig 3). In both adolescent and adult mice, cumulative energy intake over 72 hours was increased as a result of individual housing (Tables 3 and 4). During the sucrose preference test IND the sucrose preference of IND during adulthood was lower than that of SOC ($p < 0.05$), indicative of anhedonia, while diet did not significantly affect sucrose preference (preference index: IND CTR, 0.55 ± 0.02; IND WSD, 0.50 ± 0.02; SOC CTR, 0.59 ± 0.02; SOC WSD, 0.50 ± 0.02).

**Table 4. Indirect calorimetry during adulthood.**

|  | IND-CTR | IND-WSD | SOC-CTR | SOC-WSD |  |
|---|---|---|---|---|---|
|  | (*n = 25–27*) | (*n = 24*) | (*n = 11–12*) | (*n = 9*) |  |
| Energy Expenditure Light Phase |  |  |  |  |  |
| (kJ/g body weight) | 0.76 ± 0.01 | 0.65 ± 0.01 | 0.71 ± 0.01 | 0.57 ± 0.02 | a, b |
| Energy expenditure Dark Phase |  |  |  |  |  |
| (kJ/g body weight) | 0.87 ± 0.02 | 0.71 ± 0.02 | 0.84 ± 0.02 | 0.67 ± 0.02 | b |
| Cumulative energy intake (72 hour) |  |  |  |  |  |
| (kJ/ g body weight) | 5.03 ± 0.16 | 4.67 ± 0.20 | 4.55 ± 0.24 | 4.16 ± 0.27 | a |

Average Expenditure (kJ) during light (12 hr) and dark (12 hr) phase and cumulative energy intake of adult (PN106-108) mice housed individually (IND) or socially (SOC). Data are means *per cage* ± SEM;

[a] = significant effect of housing ($p < 0.05$).

[b] = significant effect of adult diet ($p < 0.05$).

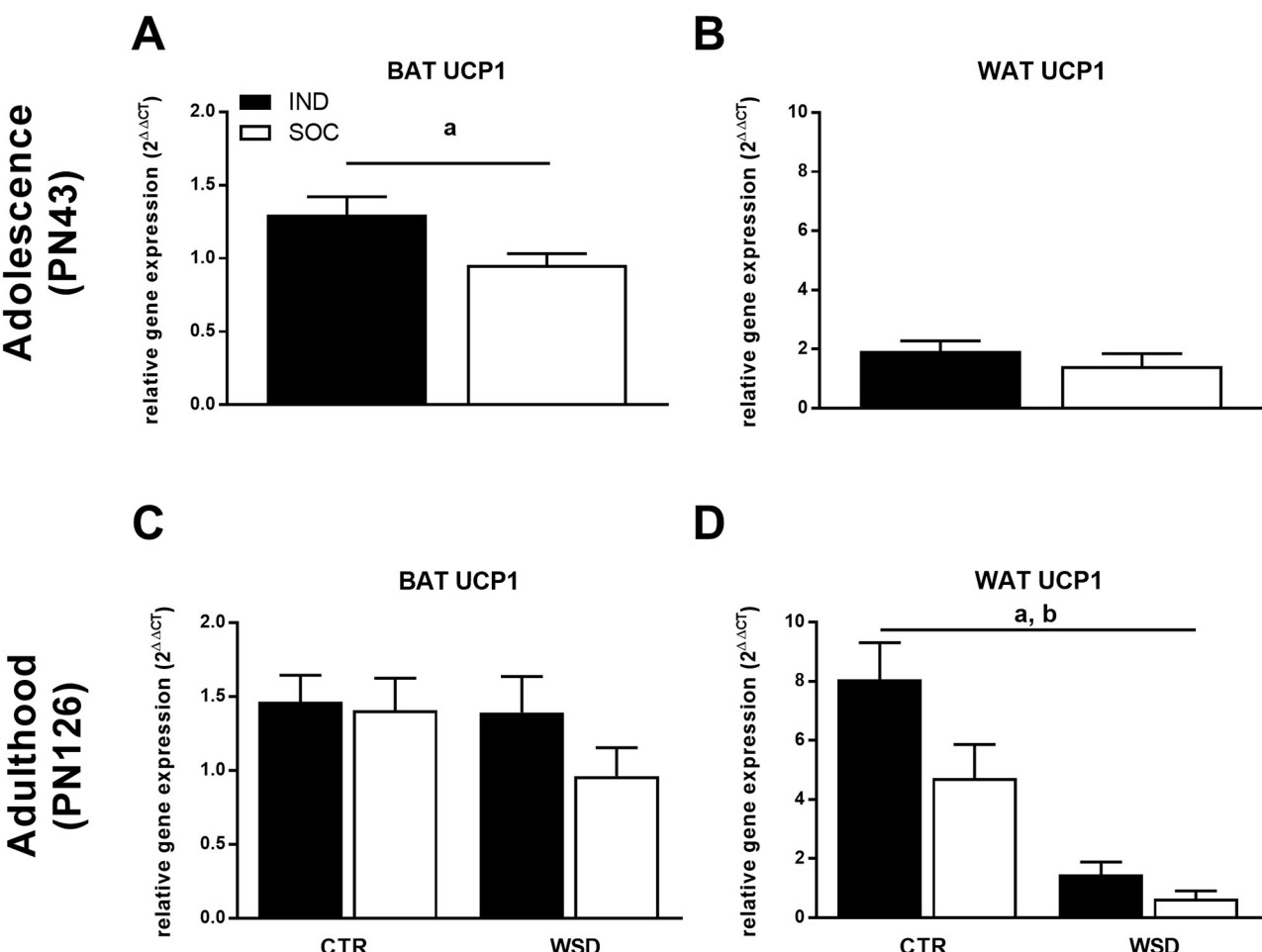

**Fig 3. Relative UCP1 mRNA expression of individually and socially housed mice.** Relative UCP1 expression in (A) interscapular BAT and (B) inguinal WAT of adolescent (PN43) mice housed individually (IND, $n = 27$) and socially (SOC, $n = 29$–$30$); (C) interscapular BAT and (D) inguinal WAT of adult (PN126) mice housed individually or socially and exposed to either control diet (CTR) or western style diet (WSD) between PN42 and PN126 (IND CTR, $n = 27$; IND WSD, $n = 18$–$24$; SOC CTR, $n = 20$–$22$; SOC WSD, $n = 15$–$17$). Data are means ± SEM; [a] = significant (main) effect of housing ($p < 0.05$); [b] = significant main effect of diet ($p < 0.05$).

## Discussion

Post-weaning individual housing reduced adolescent growth rate and predisposed animals to higher body fat accumulation, which was further exacerbated in adulthood when exposed to a moderate western style diet. These changes were observed together with increased energy intake and energy expenditure. These results indicate that post-weaning individual housing for male C57BL6j mice affects metabolic health status on short- and long term.

While others have reported no effects of post-weaning individual housing on body weight of male C57BL6j mice at 4 [34] and 6 weeks after weaning [38, 39], the current study shows that individual housing reduces growth rate earlier (i.e. up till 3 weeks) after weaning. Whereas the reduced body weight due to individual housing is transient, the reduced bone length and width persist into adulthood. Bone length in mice typically correlates to growth rate between 3 and 5 weeks of age [41], but as changes in bone mass and bone mineral density have also been reported in mice that were individually housed from adult age onwards [10, 42], reduced

mechanical loading (i.e. reduced physical interaction, play, fighting) may also contribute to the shorter bone length in individually housed mice compared to socially housed mice. Individually housed animals showed increased adipose tissue deposition which was accompanied by lower lean body mass percentage in adulthood, confirming previous observations in adult mice [34, 36]. The more obese phenotype of individually housed mice in our study was reflected in circulating metabolic hormones in particular during adulthood. In contrast to others [36], we did not observe a higher variation in fat mass in the socially compared to individually housed mice, which may have been caused by differences in experimental design. As an unstable dominant—subordinate relationship between cage mates may result in opposing patterns in body weight gain and adiposity [12, 43] we used siblings rather than unfamiliar mice for social housing to minimize this effect. In the current study, adiposity was increased as a result of post-weaning individual housing regardless of age and adult diet. Our findings contradict the results of Bartolomucci et al, who showed higher adiposity in mice due to individual housing while exposed to a high fat diet, but reduced adiposity when animals were kept on regular, low fat rodent chow diet [44]. In that study however, social isolation commenced at adult age (i.e. at 12 weeks), when mice were metabolically fully mature. In contrast, mice in the current study were individually housed from weaning onwards and experienced post-weaning growth impairments. The consequences of social isolation stress for brain function in rodents are known to be dependent on the life stage during which isolation takes place [28]. Similarly, the metabolic consequences of individual housing may also be modulated by the life stage during which individual housing takes place.

During adolescence, when increments in lean body mass [45] and bone growth [41] are highest, energy requirements are up to 3 to 4-fold higher compared to adulthood maintenance requirements [46]. Mice are social species by nature that engage in social thermoregulation to reduce energy costs for thermogenesis [47]. Individual housing under standard laboratory temperatures as applied in the current study (i.e. 21±2 °C), which is in fact below thermoneutrality for singly housed mice [24], can lead to considerable increases in energy expenditure for maintenance of normal body temperature [48–50]. This increase in thermogenesis can be reflected by increased deposition and /or activity of brown adipose tissue (BAT) [51–53] as also observed as a result of individual housing in the current study. While energy expenditure of the socially housed mice in the current study could not be determined at the individual level, the calculated values per cage suggest increased energy expenditure in IND versus SOC. Different energy partitioning in individually housed mice to support increased thermogenesis may restrict energy availability for other purposes such as growth. In line with this, rearing at lower temperatures has been reported to result in reduced (bone) growth [54].

A higher energy intake was observed as a result of individual housing. Mice are known to increase their energy intake to compensate for increased energy costs for thermogenesis [48]. A different signaling of satiety cues to the brain in the early post-weaning period may influence the structural maturation of the hypothalamic circuitry responsible for the (homeostatic) regulation of food intake [33]. Early life energy availability can thereby have permanent effects on food intake behavior. Alterations in hedonic regulation of food intake may also contribute as individual housing has been shown to modulate reward-sensitivity and depressive-like behavior [15]. In contrast to the increased energy intake in individually housed animals, we found that these animals had reduced sucrose preference during adulthood compared to socially housed animals, indicative of the fact that individual housing leads to anhedonia [55]. It has been proposed that especially the intake of palatable or high fat diet may be increased by individual housing [44]. Although increased energy intake and reduced sucrose preference index in the current study appeared to be most pronounced in individually housed animals on WSD diet, there was no significant interaction between diet and housing.

Individual housing increased adiposity at 6 weeks of age and in adulthood. The period between weaning and sexual maturation represents a critical phase for the development of white adipose tissue [30–32]. Whereas higher energy expenditure would lead to lower, rather than higher adiposity it may be hypothesized that the increased white adipose tissue deposition observed in individually versus socially housed mice in the current study is the result of an energy intake (persistently) higher than that required to support the metabolic demands. In addition, white adipose tissue maturation can be disturbed by alterations in circulating gluco-corticoids [56], which could affect (later in life) deposition. Basal CORT levels were however not affected during adolescence due to individual housing (but see below). As (home cage) physical activity levels were not monitored in the current study, a potential contribution of that factor to the differences in fat deposition observed between individually and socially housed mice in the current study cannot be excluded, although such an effect may be unlikely [10]. It remains unknown whether the increased adiposity in individually housed mice may be of functional relevance to the animal. Although it is reasonable to assume that individually housed mice would benefit from thermal insulation, white adipose tissue does not seem to have this function [57]. Increased white adipose deposition may however serve the purpose of energy storage that can be released when needed. Alternatively, increased deposition of white adipose tissue with higher thermogenic capacity (i.e. by increased browning), as suggested by higher WAT UCP-1 expression in individually compared to socially housed mice during adulthood in the current study, may facilitate overall increase of thermogenic capacity of individually housed animals without the need of further increasing BAT capacity. Such an effect may be less required in mice subjected to WSD relative to mice on the control diet, potentially due to alternative thermogenic mechanisms in an obesogenic state (8). Our data do not provide answers to what extent alterations in functioning of the HPA axis plays a role in these mechanisms. Whereas emotionally stressful experiences in rodents such as social isolation during adolescence can permanently alter HPA function (see for a review [58], this may not be reflected in higher (basal) corticosterone levels in individually compared to socially housed male mice [19, 59–61]. Indeed, basal circulating CORT levels of mice in the current study were not affected by housing conditions during adolescence and tended to be lower in individually versus socially housed mice during adulthood. Potential effects of individual housing on altered HPA responsiveness [19, 59], could however not be excluded based on the current study design. Reduced basal corticosterone levels in adult mice fed WSD confirms a blunting of basal HPA activity due to high fat diet [62].

The results of this study are of direct relevance to (metabolic) health and obesity research using mice models. The standard environmental temperature in laboratories (i.e. around 20˚C), is below thermoneutrality for mice [24]. Whereas there is no consensus on the optimal ambient temperature for mice in order to best model human physiology and disease (e.g. [50, 63–65]), there is agreement that temperatures below 21˚ are too cold for mice housed individually. Cold stress has been proposed as a factor that limits the translational value of mice models to study how environmental factors and or treatments modulate human obesity and other disease [66–68]. Whereas environmental temperature was not included as a variable in the current study, our data confirm that individual housing of mice compared to social housing at these standard temperatures alters energy balance and metabolic phenotype that may lead to even further deviation from human physiology. Individual housing may therefore be regarded as an additional limitation to the translational value of such mouse research models. Moreover, the results from this and other studies [8, 10] show that the metabolic derangements observed in individually versus socially housed mice are not always reflected in different body weights. This stresses that potential confounding influences of environment on mouse physiology may remain unnoticed when only body weight is monitored.

Different forms of (social) housing may be applied for laboratory mice depending on the study design and local regulations. In the current study social housing comprised 2 male mice (siblings) per cage, representing the lowest possible cage density under social circumstances. Moreover, all mice were kept in static cages containing wood shavings as bedding, cage enrichment (shelter) and nesting material (nestlets). Whereas not identified as a significant moderator in the systematic review and meta-analysis for rats and mice combined [8], one may speculate that the metabolic differences between individually and socially housed mice could be even more pronounced when mice are housed in larger groups (i.e. >2). A higher cage density for mice has been reported to increase within-cage temperature [48], thereby reducing the need for energy intake to support thermogenesis. Likewise, within-cage environment (e.g. cage enrichment items, amount and type of bedding and nesting materials) and ventilation systems (static vs individually ventilated cages) may further modulate the metabolic consequences of individual housing in mice as these factors influence cage temperature and can lead to behavioral and metabolic adaptations [69–74]. The potential (interaction) effects of different cage densities and cage environments on mouse health remains to be systematically evaluated in future research.

For humans, poor weight gain during the first two years of life is recognized as a risk factor for obesity and metabolic disease at later life stages [75, 76]. While we here show that post-weaning individual housing of mice also reduces early life growth rate and predisposes to (adult) obesity, there are critical differences in the underlying mechanisms that prevent individual housing to be used as a model per se for (programmed) obesity in humans. In humans, poor weight gain early in life generally results from insufficient energy intake and/or nutrient malabsorption, resulting in long lasting adaptations in energy balance regulation such as increased energy intake, lower energy expenditure and altered fat metabolism that favor a positive energy balance throughout life [77]. Rather than insufficient energy intake during critical phases of early life growth and metabolic development, individual housing in mice alters energy partitioning and increase energy expenditure.

In summary, this study shows that individual housing of male mice from weaning onwards causes substantial alterations in growth, body composition and energy balance regulation and predisposes to later in life obesity. We conclude that for (metabolic) research models with mice, the (social) housing practices should be carefully considered and regarded as a potential modulator of (metabolic) health outcomes, which may complicate the translational value of study results to the human situation.

## Methods

### Animals

All experimental procedures complied to the principles of laboratory animal care and were carried out in compliance with national legislation following the EU-Directive 2010/63/EU for the protection of animals used for scientific purposes and were approved by the ethics committee for animal experimentation (DEC-Consult, Soest, The Netherlands). All animals were kept in a controlled environment (12/12h light/dark cycle with lights on at 08:00, 21±2 °C) with *ad libitum* access to food and water, unless specified otherwise. Male C57BL/6J mice were bred in house, breeder dams and males were obtained from Charles River laboratories (Sulzfeld, Germany. After 2 weeks acclimatization, two females were introduced to a cage with a male. After 3 days, the male was removed and after 2 weeks females were individually housed and left undisturbed until birth of the litter. After birth, at PN2, litters were randomized and culled to 6 pups per dam (male:female ratio, 4:2 or 3:3). Body weight was recorded on weekly base starting at PN21 onwards.

## Housing conditions

All mice were housed in static, polycarbonate type III open cages, with bedding (Aspen wood shavings), nesting material (nestlet) and a plastic shelter (Red house; (Techniplast, Va, Italy). Directly after weaning at PN21 male offspring were moved to a different room and were randomly allocated to either of two housing conditions; housed individually (**IND**; 1 animal per cage) or housed socially with a littermate (**SOC**; 2 animals per cage). All cages were randomly placed in racks in the same room while no attempt was made to prevent visual, auditory or olfactory contact between mice in neighboring cages and / or cages elsewhere in the room. Animals remained in their respective housing conditions until sacrifice at either adolescent age (PN43) or at adult age (PN126).

## Diets

All rodent diets (Research Diet Services, Wijk bij Duurstede, the Netherlands) were semisynthetic and were based on the American Institute of Nutrition (AIN)-93 purified diets [78]. Breeder animals, dams and offspring until PN42 were kept on a AIN-93G (growth) based formulation whereas from PN42 onwards mice were subjected to a AIN-93M (maintenance) based formulation either with normal fat content (control (**CTR**); 20En% as fat) or a moderate high fat content (Western Style Diet (**WSD**); 40En% as fat) until the end of the study.

## Study design

Effects of social housing conditions on metabolic phenotype were investigated during adolescence (PN21-42) and during adulthood (PN42-126; while exposed to either CTR or WSD) using different animals. In total six groups were used in this study: two groups for collection of readouts during adolescence (individual housing, **IND**, n = 28; social housing, **SOC**, n = 30; dissection at PN43) and 4 groups for collection of readouts during adulthood (individual housing and control diet, **IND CTR**, n = 27; individual housing and Western Style diet, **IND WSD**, n = 24; social housing and control diet, **SOC CTR**, n = 24; social housing and Western Style Diet, **SOC WSD**, n = 20; dissection at PN126). One animal in the SOC AIN group was excluded from statistical analysis due to malocclusion, the animal was however kept in the study to avoid individual housing of its cage mate.

## Metabolic readouts

**Indirect calorimetry and other metabolic chamber recordings.** Individually and socially housed mice were placed in their home cage in enclosed boxes with controlled circulation for 72 hours (adolescence, PN40-42; adulthood, PN106-108). An open-circuit indirect calorimeter system allowed determination of $rO_2$ (l h$^{-1}$) and $rCO_2$ (l h$^{-1}$). The flow of inlet air was regulated by a mass flow controller (Type 5850 Brooks mass flow controller, Rijswijk, the Netherlands). After passing through the individual boxes, the air was dried (3 Å molecular sieve drying beads, Merck, Darmstadt, Germany) and the concentrations of $O_2$ and $CO_2$ (inlet and outlet air) from each individual box was measured with a paramagnetic $O_2$ analyzer (Sevomex Xentra 4100, Crowborough, UK) and an infrared $CO_2$ gas analyzer (Servomex 1440). These gas analyzers were calibrated with two gas mixtures of which $O_2$ and $CO_2$ concentrations were known. After 24 hours of acclimatization, gas exchange ($O_2$ and $CO_2$) was continuously recorded per cage. The corresponding respiratory quotient (RQ) was calculated as the ratio of $CO_2$ production and $O_2$ consumption. Energy expenditure (kJ/time unit) was calculated using the formula: energy expenditure = ([(RQ—0.70) / 0.30] x 473) + ([(1.0—RQ) / 0.30] x 439) x $VO_2$. As energy expenditure is influenced by body weight and body composition [79], all data

were adjusted for total body weight (g) of mice in the cage. In addition, data from adolescent animals were adjusted for total lean body mass (g) of mice in the cage—based on carcass LBM at PN43-. For adult animals however, this correction could not be made as metabolic chamber recordings took place between PN106-108 but carcass LBM was not determined until PN126. Total energy intake (kJ) over 72 hours in the metabolic chambers was recorded per cage by weighing the hopper with food and intake was adjusted for the total body weight (and LBM where applicable) of mice present in the cage.

**Sucrose preference test.**   To test effects of housing condition and diet on reward sensitivity, a sucrose preference test was performed over a 6-day period during adulthood, from PN74 to 80. Two bottles were present in each cage, one with tap water and the other with 0.1% sucrose (PN74 to 76) or 1% sucrose (PN77 to 80). Position of the bottles, left or right, was alternated each day. The daily consumption of water and sucrose was registered by weighing the bottles daily and preference index was calculated as average daily sucrose water (ml) consumed / total liquid intake from PN77 to 80. For socially housed animals, the values per cage were divided by 2.

**Tissue collection and body composition.**   Mice were sacrificed at adolescent age (PN43) or at adult age (PN126). On the evening prior to sacrifice mice were provided with 2 g food/ animal in the cage to induce a fasting state. The next morning, mice were anaesthetized by isoflurane inhalation which was followed by heart puncture and decapitation. Blood was collected in EDTA tubes and centrifuged at 2600 G for 10 minutes, plasma was stored at -80˚C until further analyses. Liver, WAT depots (inguinal; epididymal; retroperitoneal; perirenal) and interscapular BAT were removed from the carcass, weighed on a micro-scale and snap frozen. Length and width of the right femur were measured using a digital micro-caliper. Carcasses were dried till constant weight at 103˚C (ISO 6496–198 (E)), followed by fat extraction with petroleum ether (Boom BV, Meppel, the Netherlands) in a soxhlet apparatus. Total body fat % was calculated by (total carcass fat + weight of dissected WAT and BAT depots)/body weight (g). All tissue analyses were performed by a technician blinded to the experimental conditions.

**Plasma measurements.**   Plasma hormones corticosterone (CORT), adiponectin, leptin and insulin were analyzed in duplicate by commercial ELISA's according to manufacturer's instructions (EIA CORT kit, Arbor Assays, Michigan, USA; adiponectin, insulin and leptin, Milliplex Mouse Adipokine Multiplex, Millipore, Amsterdam, The Netherlands).

**RNA isolation and quantitative real-time PCR.**   In order to study whether the housing conditions and or adult diet affected (non-shivering) thermogenic capacity, the mRNA expression of UCP-1 was examined in BAT and inguinal WAT. RNA was isolated and cDNA was formed using commercial kits according to manufacturer's instructions (NucleoSpin miRNAs kit; Macherey-Nagel, Düren, Germany; iScript cDNA synthesis kit, Bio-Rad, Veenendaal, Netherlands). Quantity and chemical purity of RNA and cDNA were assessed using Nanodrop 2000 spectrophotometer (Thermo Fisher Scientific). mRNA expression was measured using real time polymerase chain reaction (RT-PCR) according to a protocol previously described [80]. The RT-PCR was performed in triplicates by using validated UCP-1 primers (forward primer `CAAAAACAGAAGGATTGCCGAAA`, reverse primer `TCTTGGACTGAGTCGTAGAGG`), with housekeeping gene RPL13A, SYBR green and the CFX96 qPCR (Bio-Rad, Veenendaal, Netherlands). UCP-1 expression values were normalized to non-targeted RPL13A expression levels within the same sample to determine deltaCt (deltaCt = Cq gene–Cq). The deltaCt values for each replicate was then exponentially transformed to deltaCt expression (Mean deltaCt expression; [2^(Cq gene—Cq RPL13A)]). DeltaCt expression values were finally averaged for each triplicate. LinRegPCR (version 12.15, 2011) was used to calculate PCR efficiencies for each sample.

## Statistical analysis

Statistical analyses were performed using SPSS 22.2 (IBM Software). Effects of housing conditions on growth during adolescence (PN21-42) were analyzed by analysis of variance (ANOVA) with repeated measures (PN as the repeated measures). Body weight gain during adulthood (PN42-126) was analyzed by two-way ANOVA with repeated measures (adult diet x housing conditions with PN as the repeated measures). All other parameters were analyzed by one-way ANOVA (housing conditions, adolescence) or two-way ANOVA (housing conditions x diet, adulthood), significant interaction effects were followed by Tukey's post hoc test. All data are presented as mean ± SEM and considered significantly different when $p < 0.05$.

## Supporting information

**S1 Table. Absolute bodyweight of adult mice housed individually or socially and exposed to either control diet (CTR) or Western Style diet (WSD) between PN42 and PN126.** Data are means ± SEM; [a] = significant main effect of housing ($p < 0.05$); [a] = trend ($0.05 < p < 0.1$); [b] = significant main effect of diet ($p < 0.05$); [c] = significant interaction housing x diet ($p < 0.05$); [c] = trend ($0.05 < p < 0.1$); [d] = significantly different from IND-CTR ($p < 0.05$); [e] = significantly different from IND-WSD ($p < 0.05$); [f] = significantly different from SOC-CTR ($p < 0.05$). (DOCX)

## Author Contributions

**Conceptualization:** Lidewij Schipper, Gertjan van Dijk.

**Formal analysis:** Lidewij Schipper, Steffen van Heijningen, Giorgio Karapetsas.

**Investigation:** Lidewij Schipper, Steffen van Heijningen, Giorgio Karapetsas, Gertjan van Dijk.

**Project administration:** Lidewij Schipper.

**Supervision:** Gertjan van Dijk.

**Writing – original draft:** Lidewij Schipper, Steffen van Heijningen.

**Writing – review & editing:** Eline M. van der Beek, Gertjan van Dijk.

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
