## [Decision Letter · Decision Letter 0]

3 Jan 2020

PONE-D-19-30739

Individual housing of male C57BL/6J mice after weaning impairs growth and predisposes for obesity

PLOS ONE

Dear Dr. Schipper,

Thank you for submitting your manuscript to PLOS ONE. After careful consideration, we feel that it has merit but does not fully meet PLOS ONE’s publication criteria as it currently stands. Therefore, we invite you to submit a revised version of the manuscript that addresses the points raised during the review process.

We would appreciate receiving your revised manuscript by Feb 17 2020 11:59PM. To enhance the reproducibility of your results, we recommend that if applicable you deposit your laboratory protocols in protocols.io, where a protocol can be assigned its own identifier (DOI) such that it can be cited independently in the future. For instructions see: http://journals.plos.org/plosone/s/submission-guidelines#loc-laboratory-protocols

We look forward to receiving your revised manuscript.

Kind regards,

Kathleen R. Pritchett-Corning, D.V.M.

Academic Editor

PLOS ONE

Journal Requirements:

3. Thank you for including your competing interests statement; "LS and EMvdB are employed by Danone Nutricia Research. SvH, GK and GvD declare no conflict of interest."

We note that one or more of the authors are employed by a commercial company: Danone Nutricia Research

Reviewers' comments:

Reviewer's Responses to Questions

**Comments to the Author**

1. Is the manuscript technically sound, and do the data support the conclusions?

Reviewer #1: Yes

Reviewer #2: Yes

2. Has the statistical analysis been performed appropriately and rigorously? 

Reviewer #1: Yes

Reviewer #2: No

3. Have the authors made all data underlying the findings in their manuscript fully available?

Reviewer #1: Yes

Reviewer #2: Yes

4. Is the manuscript presented in an intelligible fashion and written in standard English?

Reviewer #1: Yes

Reviewer #2: Yes

5. Review Comments to the Author

Reviewer #1: Overall this study successfully demonstrated the effects of individual housing of male mice on body weight and composition during adolescence and further in adulthood with standard or increased caloric intake diets. This is an important aspect of the current interest in obesity and in particular the use of rodent models. Since individual housing is a common approach for rodent studies the current findings are very pertinent to this, and other, area(s) of research using mice.

Despite this, the authors have not made any comment on their decision to use pairs of mice as the social housing comparison. Mice are very commonly housed in larger groups, or even as trios, as has been the case in a number of the studies referenced in this manuscript. Certainly the authors demonstrated significant differences between the two groups of differently housed mice, however, it would have more applicability and relevance if the “control” group where a more common grouping. The general cage temperature is likely to be even higher in multiply-housed mice than when in pairs, indicating that housing mice in pairs may also be a form of chronic stress.

1. Abstract

The abstract lacks any sortt of introduction and starts with the results which provides no context for conducting the study. This is partly resolved by the last sentence of the introduction, which should probably be the first sentence.

2. Results

Most of the results are clearly and concisely presented, however there were a few that need further or alternate explanation. For example, it is unclear why, in Figure 1A the results are expressed as absolute body weight while in Figure 1B they are weight gain. This diminishes the effect of reduced weight gain in days PN21-42 in the IND mice and leads one to believe that at the start of the different diets the mice were all the same weight. It also means the reader cannot see at what age the IND mice become heavier that the SOC mice.

In Figures 2 and 3 the authors have used different y-axis scales for the same parameter for the adolescent and adult mice and in some instances have not started at zero. This approach certainly highlights the differences between groups within each individual figure but makes comparisons between figures difficult. Since most differences are statistically significant there is no reason to “inflate” the results by manipulating the figures to emphasize statistical differences.

In Table 2 for absolute body weights the authors have included a superscript for a “trend” according to the legend for Table 3 which does not appear in the Table 2 legend. Further, the text (line 106) states that the body weight was greater but not statistically for IND mice regardless of diet, yet the table implies otherwise. So it is unclear what the statistical significance of the difference in these groups of mice really is.

Tables 3 and 4 which present the same data for adolescent and adult mice but otherwise differ in that for the adolescent mice the energy expenditure is expressed as both kJ/g body weight and kJ/g lean body mass. There is no explanation for this and the only apparent reason is that it provides another “significant” result at the p<0.05 level when otherwise it would only show a "trend".

3. Methods

For the most part the methods are clearly written although some aspects are lacking in details, making replication of the experiments difficult to achieve. For instance, it is unclear whether the caging used was individually-ventilated or open. The technique used for the indirect calorimetry is especially vague in regard which equipment was used and from which manufacturer it was purchased. There is indication whether the plasma hormone analyses were done in either duplicate or triplicate, which presumably they were.

Reviewer #2: The study investigate metabolic performance of mice in response to individual or collective housing.

The work is rigorous, well conducted and data are presented clearly. The paper is very well written and pleasant to read.

Criticisms: the paper seems to be incremental only and originality does not clearly appear in regards to previous works on the topic. Please indicate previous findings in last paragraphs of the introduction and indicate the interest of this new study. Also include additional references in the discussion to compare with these findings.

6. PLOS authors have the option to publish the peer review history of their article (what does this mean?). If published, this will include your full peer review and any attached files.

Reviewer #1: Yes: Anthony Nicholson, School of Animal & Veterinary Sciences, University of Adelaide, South Australia

Reviewer #2: Yes: Alexandre Benani

---

## [Author Response · Author response to Decision Letter 0]

13 Feb 2020

Response to editors comments: 

General: “Authors have investigated metabolic performance and vulnerability to diet induced obesity in mice in response to individual or collective housing. Findings are clear and not over-interpreted. However novelty/originality is not clear and the paper seems to be incremental only. To address this point, I suggest inserting a mini-review based on previous papers in the introduction section, with a statement on what we know and what we do not know actually, just before giving the main objective of the present work.”

Response: We thank the reviewer for this suggestion. A few lines have been added to the introduction where current (gaps in) knowledge about metabolic consequences of post-weaning individual housing for mice have been addressed specifically.

Comment 1: “Please indicate issues from previous works on the same topic in the introduction section and state questions that still remain, which will improve interest for the present work. For instance, studies that are already cited in the discussion should be included in this introduction, to reveal lacks or discordance on the effect of housing on energy homeostasis. Studies already cited in the discussion are: Tsuduki et al. 2012; Bibancos et al. 2007; Lopez et al. 2015; Shin et al 2018; Nagy et al. 2002; Moles et al. 2006.”

Response: In line with the previous comment of this reviewer, we have indeed now used these and some additional references in the introduction where appropriate and describe what knowledge is still lacking.

Comment 2: “Others studies not mentioned in the first version of this paper should appear here as well:

Breslin et al. Laboratory Animals 44, 2010; Guo et al. Prog Neuropsychopharmacol Biol Psychiatry 28, 2004; Nonogaki, Biochem Biophys Res Commun 378, 2009; Nonogaki, Endocrinology 2007; for instance’’ 

Response: These references have been added in the introduction when appropriate. 

Comment 3: “More recent papers on thermoneutrality in mice and its impact on DIO models should be included. See Keijer et al. Mol Metab 25, 2019, and Fischer et al. Mol Metab 7, 2018, and related comments in 2019”

Response: A few lines have been added to the discussion section about the current debate on what is the optimal environmental temperature for mice to mimic human physiology and disease. The references suggested by the reviewer have been included. In addition, three (recent) review articles (i.e. Karp 2012; Ganeshan 2017; Hankenson 2018) addressing translational value of mouse models in relation to environmental temperature have been cited. 

Comment 4 “For statistical analysis, post hoc tests should be performed after ANOVA”.

Response: Post hoc tests have been performed after ANOVA upon identification of a significant housing*diet interaction effect. This has now been described in the methods section. 

Additional comment that was raised by the editor in the decision letter: “We note that you have included the phrase “data not shown” in your manuscript [….]. Please add a citation to support this phrase or upload the data that corresponds with these findings to a stable repository” 

Response: The phrase “data not shown” has been removed. The data have now been added as supplementary table 1. 

 

Response to reviewers comments: 

Reviewer #1

General: “Overall this study successfully demonstrated the effects of individual housing of male mice on body weight and composition during adolescence and further in adulthood with standard or increased caloric intake diets. This is an important aspect of the current interest in obesity and in particular the use of rodent models. Since individual housing is a common approach for rodent studies the current findings are very pertinent to this, and other, area(s) of research using mice. 

Despite this, the authors have not made any comment on their decision to use pairs of mice as the social housing comparison. Mice are very commonly housed in larger groups, or even as trios, as has been the case in a number of the studies referenced in this manuscript. Certainly the authors demonstrated significant differences between the two groups of differently housed mice, however, it would have more applicability and relevance if the “control” group where a more common grouping. The general cage temperature is likely to be even higher in multiply-housed mice than when in pairs, indicating that housing mice in pairs may also be a form of chronic stress”.

Response: We thank the reviewer for noting the relevance of the current work to (obesity related) rodent studies in which individual housing is common practice. 

The reviewer rightfully points out that large variety exists in the form in which social housing is applied. For instance, in our lab, pair housing is the default form of social housing for male mice, but other labs may have different preferences for cage density depending on husbandry, previous experience and or (local) regulations. The objective of the current study was to evaluate effects of individual vs social housing in mice. Two mice per cage represents the minimum number of animals per cage required for ‘social housing’. One may speculate that differences between individual and group housed mice (>2 per cage) may even be more pronounced as temperature within the cage may increase with density. However, comparing effects of different types of social housing (i.e. densities) on metabolic health parameters in mice was beyond the scope of the current study. We have added a paragraph to the discussion (lines 284-298) to elaborate on potential effects of cage density and other within-cage environmental factors (which may also interact with cage density) to modify metabolic outcomes. 

Abstract Comment 1. “The abstract lacks any sort of introduction and starts with the results which provides no context for conducting the study. This is partly resolved by the last sentence of the introduction, which should probably be the first sentence”

Response: We have followed the reviewer’s suggestion and have now started the abstract with an introduction, providing more context to the topic (using also the former last sentence of the abstract). 

Results Comment 1: “Most of the results are clearly and concisely presented, however there were a few that need further or alternate explanation. For example, it is unclear why, in Figure 1A the results are expressed as absolute body weight while in Figure 1B they are weight gain. This diminishes the effect of reduced weight gain in days PN21-42 in the IND mice and leads one to believe that at the start of the different diets the mice were all the same weight. It also means the reader cannot see at what age the IND mice become heavier that the SOC mice”

Response: Figure 1A describes the effects of housing conditions on weight gain between PN21 and 43. Groups are similar at baseline (PN21); therefore, the data are expressed as absolute body weight. Figure 1 B describes effects of housing conditions and diet intervention between PN42 and PN126. PN42 is considered as baseline for the adult group as it is the start of a new diet (WSD or AIN-93-M). As described in the figure caption 1B and in the accompanying text in the results section, the (absolute) bodyweight of animals at baseline (PN42) was significantly different between socially and individually housed groups. Unequal characteristics at baseline may be considered as a source of bias in animal studies (See for example Hooijmans et al. BMC Medical Research Methodology 2014, 14:43; doi: 10.1186/1471-2288-14-43). To correct for unequal distribution of groups at baseline the body weight is expressed as gained weight from P42 onwards in figure 1B. 

Several changes have been made to the text in manuscript and figures to avoid further confusion on this matter:

- In the objective (in introduction) it is stated that adult body weight gain (rather than growth) has been evaluated in the current study. 

- The title of figure 1B has been changed to “adult gained bodyweight” rather than “adult growth” to better reflect the data that is presented in the graph. In the figure caption several textual changes were made as well to emphasize the differences in absolute body weight at baseline.

- The text in the results section has been reordered in such a way that the absolute body weight at PN42 is described earlier than the gained body weight.

Furthermore, the absolute body weight of the groups at PN42, 60, 80, 100 and 126 has been added as supplementary material (supplementary table 1). 

Results Comment 2: “In Figures 2 and 3 the authors have used different y-axis scales for the same parameter for the adolescent and adult mice and in some instances have not started at zero. This approach certainly highlights the differences between groups within each individual figure but makes comparisons between figures difficult. Since most differences are statistically significant there is no reason to “inflate” the results by manipulating the figures to emphasize statistical differences”.

Response: Y-axis scales for the same parameters have now been aligned between the adolescent and adult mice and start at zero where possible. For some parameters (e.g. LBM and femur length) however Y-axis breaks have been implemented as the reader would otherwise not be able to tell from the individual figures how groups are different from each other. 

Results Comment 3: “In Table 2 for absolute body weights the authors have included a superscript for a “trend” according to the legend for Table 3 which does not appear in the Table 2 legend. Further, the text (line 106) states that the body weight was greater but not statistically for IND mice regardless of diet, yet the table implies otherwise. So it is unclear what the statistical significance of the difference in these groups of mice really is.”

Response We thank the reviewer for noting this error. As described in the text, the difference between social and individually housed animals was not significant but a trend was found. The superscript for trend in the table is therefore correct, however the figure legend was incomplete. We have adapted the figure legend accordingly.

Results Comment 4: “Tables 3 and 4 which present the same data for adolescent and adult mice but otherwise differ in that for the adolescent mice the energy expenditure is expressed as both kJ/g body weight and kJ/g lean body mass. There is no explanation for this and the only apparent reason is that it provides another “significant” result at the p<0.05 level when otherwise it would only show a "trend".

Response: Energy expenditure can be influenced by bodyweight and LBM (see line 369, reference Tschop 2011) and if possible, data should be adjusted for these parameters. For adolescent mice in the current study this correction was appropriate as the LBM was determined within 24 hours after the metabolic chamber recordings. For adult animals however, we were unable to make this correction as metabolic chamber recordings took place at PN106-108, but the LBM was not determined until PN126; > 2 weeks later. The LBM might have changed in the period in between measurements and correction for LBM was therefore considered inappropriate. The different approach used for adolescent and adult animals has now been made clearer in the method section (see line 371-373). 

Methods Comment 1:”For the most part the methods are clearly written although some aspects are lacking in details, making replication of the experiments difficult to achieve. For instance, it is unclear whether the caging used was individually-ventilated or open. The technique used for the indirect calorimetry is especially vague in regard which equipment was used and from which manufacturer it was purchased. There is indication whether the plasma hormone analyses were done in either duplicate or triplicate, which presumably they were”.

Response: We have followed the suggestion of the reviewer and have provided the missing details in the methods section of the manuscript.

 

Response to reviewers comments: 

Reviewer #2

General comment: “The study investigate metabolic performance of mice in response to individual or collective housing. The work is rigorous, well conducted and data are presented clearly. The paper is very well written and pleasant to read.

Criticisms: the paper seems to be incremental only and originality does not clearly appear in regards to previous works on the topic. Please indicate previous findings in last paragraphs of the introduction and indicate the interest of this new study. Also include additional references in the discussion to compare with these findings.”

Response: We thank the reviewer for his/her complements on the quality of the work and the readability of the paper. As suggested by this reviewer and the editor, we have now described the previous findings of other researchers using a similar model and have pointed out what was still not known. Additional references relevant to the model and results of the current study have been added at several places in the introduction and discussion.

---

## [Decision Letter · Decision Letter 1]

24 Apr 2020

Individual housing of male C57BL/6J mice after weaning impairs growth and predisposes for obesity

PONE-D-19-30739R1

Dear Dr. Schipper,

We are pleased to inform you that your manuscript has been judged scientifically suitable for publication and will be formally accepted for publication once it complies with all outstanding technical requirements.

I only noted one minor wording matter in the abstract, which I recommend to change: The term "confounder" does not seem to fit here. To my understanding, this would be a factor that confounds a difference between two study conditions. Here, I feel that something like "... housing conditions could influence ... the outcomes" would be preferable.

With kind regards,

Clemens Fürnsinn, Ph.D.

Academic Editor

PLOS ONE

Additional Editor Comments (optional):

I noted one minor wording matter in the abstract, which I recommend to change: The term "confounder" does not seem to fit here. To my understanding, this would be a factor that confounds a difference observed between two study conditions. Here, I feel that something like "... housing conditions could influence ... the outcomes" would be preferable.

Reviewers' comments:

Reviewer's Responses to Questions

**Comments to the Author**

1. If the authors have adequately addressed your comments raised in a previous round of review and you feel that this manuscript is now acceptable for publication, you may indicate that here to bypass the “Comments to the Author” section, enter your conflict of interest statement in the “Confidential to Editor” section, and submit your "Accept" recommendation.

Reviewer #2: All comments have been addressed

2. Is the manuscript technically sound, and do the data support the conclusions?

Reviewer #2: Yes

3. Has the statistical analysis been performed appropriately and rigorously? 

Reviewer #2: Yes

4. Have the authors made all data underlying the findings in their manuscript fully available?

Reviewer #2: Yes

5. Is the manuscript presented in an intelligible fashion and written in standard English?

Reviewer #2: Yes

6. Review Comments to the Author

Reviewer #2: (No Response)

7. PLOS authors have the option to publish the peer review history of their article (what does this mean?). If published, this will include your full peer review and any attached files.

Reviewer #2: Yes: Alexandre Benani

---

## [Editor Report · Acceptance letter]

12 May 2020

PONE-D-19-30739R1 

Individual housing of male C57BL/6J mice after weaning impairs growth and predisposes for obesity 

Dear Dr. Schipper:

I am pleased to inform you that your manuscript has been deemed suitable for publication in PLOS ONE. Congratulations! Your manuscript is now with our production department. 

With kind regards,

on behalf of

Prof. Dr. Clemens Fürnsinn 

Academic Editor

PLOS ONE